# Is Memorization *Actually* Necessary for Generalization

## Abstract

Deep learning models are known for their ability to memorize training data. While memorization is often linked to risks such as privacy leakage and poor robustness, a highly influential claim by Feldman & Zhang (2020) argues that memorization is actually *necessary* for generalization. Their conclusion is based on the observation that removing points with high memorization scores reduces test accuracy. Upon closer inspection of their work, we uncover four critical flaws in the underlying methodology: **(1) sampling bias** in their approximation algorithm inflates memorization scores; **(2) high false positive rate** in their definition of memorization, leads to misclassification of non-memorized points as memorized; **(3) unprincipled thresholding** that results in an ill-posed problem; and **(4) data leakage** skews the test accuracy results. To address these limitations, we introduce a modifications for correctly identifying and evaluating memorization, including higher sampling rates, modifying the original memorization definition to reduce the false positive rates, proposing a method to identify a principled score threshold, and employing test datasets especially designed to avoid data leakage. Having accounted for these errors, our results show that, in contradiction to the original work, removing truly memorized points does not cause a drop in accuracy, and in most cases, improves test performance. These findings call into question the necessity of memorization in deep learning and highlight the importance of mitigating its risks.

## 1 Introduction

Deep learning models possess a remarkable ability to *memorize* data. They can fit arbitrary inputs, even when they lie far outside the natural data distribution. For instance, image classifiers can be trained to associate images of Gaussian noise with the label cat (Zhang et al., 2017; Arpit et al., 2017; Stephenson et al., 2021). Similarly, LLMs can fit random strings (such as addresses or even secret keys) and regurgitate them when properly prompted Carlini et al. (2019). This phenomenon is only possible because of the model's capacity to memorize training data.

At the heart of current literature lies a key tension. On one hand, a substantial body of work has documented the downsides of memorization. They highlight its role in leakage of private data, model stealing attacks, increased model bias, poor robustness to distribution shifts, and susceptibility to adversarial examples (Bayat et al., 2024; Black & Fredrikson, 2021; Usynin et al., 2024; Carlini et al., 2022; 2023; 2021; Somepalli et al., 2023; Ye et al., 2022; Salem et al., 2018; Jayaraman et al., 2020; Shokri et al., 2017; Long et al., 2020). On the other hand, a highly cited result by Feldman & Zhang (2020) claims that memorization is not just incidental but in fact *necessary* for achieving strong generalization in deep networks. This raises an important question: is memorization a necessary evil?

Intuitively, memorization and generalization appear to be at odds. Generalization relies on uncovering patterns that apply to unseen data, while memorization involves storing specific training examples, offering no obvious utility for unseen inputs. However, Feldman & Zhang (2020) argue otherwise. They claim that removing data points with high memorization scores leads to a measurable drop in test accuracy, thereby suggesting that memorization supports generalization.

To make this argument, the authors introduce an approximation algorithm for estimating how much a point is memorized. They then show that removing the most memorized points from the training

set decreases test performance. However, a closer inspection reveals several critical flaws in this approach—issues that undermine the core conclusions of the paper:

1. **Sampling Bias:** The approximation algorithm proposed by Feldman (2020) introduces sampling bias, resulting in a systematic overestimation of memorization scores (error as high as 10%). This leads to non-memorized points being misclassified as memorized. (Section 3.1)

2. **High False Positive Rate:** The memorization scoring definition is never evaluated against a baseline. We show that it suffers from a high false positive rate (as high as 60% at 95% True Positive Rate), failing to reliably distinguish truly memorized points from naturally learnable ones. (Section 3.2)

3. **Unprincipled Thresholding:** No clear statistical criterion is given for choosing a threshold to distinguish memorized from non-memorized points i.e., the problem is ill-posed. Since the conclusions change based on this arbitrary threshold, the findings lack robustness. (Section 3.3)

4. **Data Leakage:** Some points labeled as highly memorized have near-duplicates in the test set. Thus, the observed test accuracy drop upon removing these points may stem from test-train overlap rather than a causal role of memorization in generalization. (Section 3.4)

**Re-evaluating the Role of Memorization.** Motivated by these issues, we introduce principled fixes. We use high data sample rates to reduce bias, prose a memorization definition based on fractional difference to reduce false positive, provide a mechanism for statistically grounding the thresholds, and employ test datasets specifically designed to avoid leakage. *Our experiments show that removing memorized at worst does not impact accuracy, and at best, increases test accuracy.* These results challenge the notion that memorization is necessary for generalization, and in fact show that their presence harms generalization. Given its well documented downsides, we argue that memorization should not be viewed as a desirable feature in deep learning models, but rather as a behavior to be mitigated.

## 2 Understanding Feldman & Zhang (2020)

Having gone over how different factors influence memorization, we describe in detail the original work of Feldman & Zhang (2020). Our primary goal is to evaluate their methodology, recommend experimental fixes, and consequently, reassess their findings. To that end, we describe how they 1) define memorization, 2) approximate memorization scores, and 3) quantify marginal utility.

### 2.1 Defining Memorization

Feldman & Zhang (2020) define a memorized point as one having high self-influence (i.e., a point that is predicted correctly only when present in the training data).

Specifically, consider a training set $S = ((x_1, y_1)...(x_n, y_n))$ and a point $x_i$ in the training set $S$. The memorization score is the difference in prediction accuracy between when the point $x_i$ is present in the training data ($h \leftarrow A(S)$) and when $x_i$ is absent ($h \leftarrow A(S^{\backslash i})$). Here, ($h \leftarrow A(S)$) means that models $h$ were trained on dataset $S$ using algorithm $A$. The *absolute* difference is calculated using:

$$\mathbf{Pr}_{h \leftarrow A(S)}[h(x_i) = y_i] - \mathbf{Pr}_{h \leftarrow A(S^{\backslash i})}[h(x_i) = y_i] \qquad (1)$$

The definition captures the intuition that a point $x_i$ has been memorized if its prediction changes significantly when it is removed from the dataset. We include Table 2 for reference on the symbols used throughout the paper.

For example, consider training 1000 instances each of the models $h \leftarrow A(S)$ and $h \leftarrow A(S^{\backslash i})$. If the correct classification rate for $x_i$ when it $h \leftarrow A(S)$ is around 90% (i.e., 900 out of the 1000 instances classified the point correctly). However, it falls significantly when $h \leftarrow A(S^{\backslash i})$ to 25% (i.e., 250 out of the 1000 instances classified the point correctly). Due to the significant drop in self accuracy, this point has a high self-influence, and therefore, a high memorization score, specifically of $90\% - 25\% = 65\%$. This means that $x_i$ is far more likely be classified correctly when it is present

in the training data. In contrast, if there is no significant change in the classification rate, then it has a low memorization score. In this case, $x_i$ will likely be classified correctly, whether or not it is present in the train set. As a result, the memorization score of a given point will be inversely proportional to its sub-population size: the larger the sub-population the smaller the memorization score. In the case of our hypothetical cat dataset, the pink cat (singleton) will have the highest memorization score, followed by the black cats (small-subpopulation) and the white cats (large sub-population).

Having defined memorization, the next step is to develop a methodology to identify memorized points from a dataset. A point is considered memorized based on its memorization score, calculated using Equation 1. The most *precise* way to compute this score is via the classic leave-one-out experiment. Here, we remove a single point from the training dataset, retrain the model on the remaining data, and test to see if the removed point is correctly classified. We have to run this experiment on all the points in the dataset to get the memorization score for each. Additionally, we have to repeat this model training process, for each point, multiple times to account for randomness introduced during training (e.g., the varying initialization, GPU randomness, etc.). Specifically, this would require training hundreds models for every point in the data. Considering data sets contain tens of thousands of points, this would require training millions of models. Therefore, running this experiment over a large dataset and model will require a large amount of resources and is therefore, computationally intractable.

## 2.2 Approximation Algorithm and Marginal Utility:

To overcome this limitation, Feldman & Zhang (2020) propose a method to *approximate* the memorization scores. To that end, the authors employ the following steps. **Step 1. Sampling:** Instead of removing one point at a time, the authors randomly sample a fraction $r$ of the points from the training set (originally of size $n$) and leave the remaining points out of training. The number of points used in training is then $m = r \cdot n, 0 \leq r \leq 1$. In Feldman & Zhang (2020) the authors use $r = 0.7$ for their experiments. **Step 2. Training:** Each sample of points is then used to train an individual model. As a result, a random point $x_i$ will be present in approximately $k \cdot r$ of the total trained models and will be absent from $k \cdot (1 - r)$ of them. **Step 3. Memorization Score Calculation:** The memorization scores are aggregated the results over both sets of models using Equation 1. **Step 4. Thresholding:** All the points that have a higher memorization score than some predetermined threshold are said to be memorized. **Step 5. Marginal Utility:** Having identified the memorized points, the authors now calculate their marginal utility (i.e., their impact on test accuracy).

## 3 Feldman & Zhang (2020) Limitations

Having outlined how the original authors defined and approximated memorization, we now turn to examining the various sources of error at each of the steps. In each case, we demonstrate the source of error and how to fix it. Our goal is to (1) demonstrate how these errors undermine the central conclusion about the relationship between memorization and generalization and (2) provide solutions to fix these errors. Our experiments vary over different datasets (CIFAR-10/100, and Imagenet), models (MobileNet, VGG19, Resnet18, and Resnet50), and different training parameters. We use the same training setup as Feldman (2020), with details in Appendix A. We train approximately 50 models for each dataset model pair. For succinctness, we present the RESNET-50 results in the main body of the work and leave the rest in the Appendix.

### 3.1 Sampling Bias leads to high approximation error of Memorization Scores

Step 1 in Feldman & Zhang (2020)'s approximation algorithm (Section 2.2) is to take random samples of the data in order to train the models. Here, the choice of the sampling rate hyperparameter can significantly affect the accuracy of memorization scores. An incorrect choice can lead to sampling bias, resulting in a high approximation errors. Specifically, using **lower sampling rates**, i.e., training on a small subset of the data, can introduce significant approximation error due to overfitting. This is because there are not enough points for the model to learn the over-arching pattern. Instead, the model will memorize the few points it sees during training. As a result, both memorized and non-memorized points will be marked as memorized. This does not mean that these points will be memorized by model when trained on the full dataset, but are an artifact of a low sample rate.

In contrast, **higher sampling rates**, those approaching the leave-one-out (i.e., evaluating memorization by only removing one point at a time, the gold standard for memorization calculation Feldman (2020)), will yield more reliable results. In the leave-one-out (LOO) setup, each model is trained on the entire dataset except for one held-out point, which is then evaluated. This ensures that any correct prediction for the held-out point is due to generalization from the remaining data, rather than direct memorization of its label Vapnik (1999) . Therefore, training at high sampling rates, i.e., training on large subsets of the data, will lead to more accurate results.

To empirically demonstrate the impact of sampling rate, we train ensembles of models using a range of sampling rates from 0.1 to 0.5. Since true LOO evaluation is computationally infeasible in our setting, we use a very high sampling rate (close to 0.95) as a practical baseline, which closely approximates LOO behavior. We then quantify the approximation error by comparing memorization scores with the baseline. Figure 1 and 6 show the results our experiments. We can see that across every model and dataset, as sampling rate increases, approximation error decreases. This analysis illustrates how lower sampling rates introduce significant bias, resulting in unreliable memorization measurements. Our findings provide empirical validation for the presence of sampling bias, and therefore, necessitate the the use of high rates of memorization calculation.

Unfortunately, the original work did not take this into account and employed a sampling rate of 0.7. On its own, using a this sample rate does not necessarily disprove the findings of the original work. However, this rate does produce substantial approximation error in the memorization scores. As a result, points identified as highly memorized at this sampling rate may not in reality might not have been memorized. As a result, the drop an accuracy after removing them might lead to the conclusion that memorization in needed for generalization. However, this would be flawed as the removed points contained non-memorized data as well. A precise experimental setup should only remove memorized points without removing the non-memorized data.

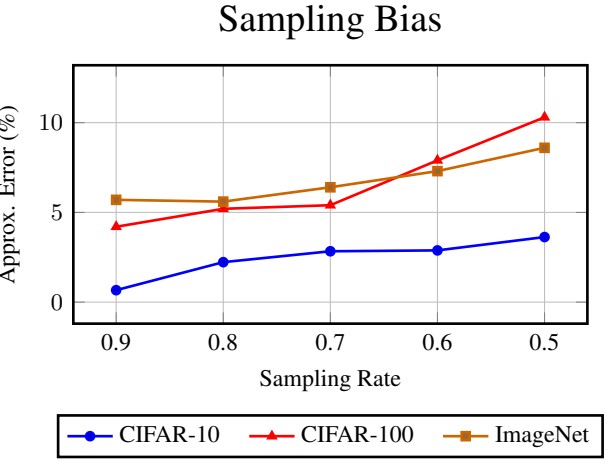

Figure 1: Lower the sampling rate leads to higher the approximation errors.

Proposed Solution: To ensure trustworthy conclusions, an accurate study of memorization vs generalization should use a sampling rate as close to leave-one-out as computational constraints permit.

### 3.2 THE ORIGINAL DEFINITION HAS HIGH FALSE POSITIVE RATES

Having trained the models from the sampled data, Feldman & Zhang (2020), Step 3 of the approximation algorithm (Section 2.2) involves calculating the memorization scores using Equation 1. A good memorization definition should distinguish between points that are truly memorized (e.g., outliers) and those that generalize well (e.g., inliers). However, we hypothesize that the simple difference score defined in Equation 1 does not do so and leads to a high false positive rate when identifying memorized points.

Consider an outlier point $x_o$ that is correctly classified by 25% of models when it is present in the training set, and by 0% of models when it is absent. According to the memorization score, its value is $25\% - 0\% = 25\%$. Now, consider an inlier point $x_i$ that generalizes well across models: it is correctly classified 100% of the time when present, and 75% of the time when absent. Its memorization score is also $100\% - 75\% = 25\%$.

Despite their fundamentally different behaviors, both points receive the same memorization score. This contradicts the intuition that $x_o$, which is *only* learned when explicitly trained on, should be considered more memorized than $x_i$, which is consistently learned even without training exposure.

As a result, the score fails to meaningfully separate memorized points from generalizable ones. This results in a high false positive rate and undermines the reliability of the conclusions.

Instead of using the simple difference, as in Equation 1, we propose using *fractional* difference:

$$\frac{\mathbf{Pr}_{h\leftarrow A(S)}[h(x_i) = y_i] - \mathbf{Pr}_{h\leftarrow A(S\setminus i)}[h(x_i) = y_i]}{\mathbf{Pr}_{h\leftarrow A(S)}[h(x_i) = y_i] + \mathbf{Pr}_{h\leftarrow A(S\setminus i)}[h(x_i) = y_i]} \tag{2}$$

This updated scoring function[1] assigns the outlier $x_o$ a higher memorization score of $\frac{25\% - 0\%}{25\% + 0\%} = 100\%$, while the inlier $x_i$ receives a lower score of $\frac{100\% - 75\%}{100\% + 75\%} \approx 14.3\%$. This ensures that $x_o$ and $x_i$ are distinguishable under the revised definition, addressing the ambiguity in the original formulation.

Now, we empirically evaluate both the original and updated definitions in their ability to identify memorized points. To do so, we use randomly labeled points as a baseline for memorized points, similar to prior work Arpit et al. (2017); Carlini et al. (2019); Chatterjee (2018; 2020). These points can only be correctly classified via memorization, as there is no underlying structure to generalize. The definition that makes fewer mistakes in distinguishing these memorized random-label points from natural examples is considered more effective. We quantify detection performance by randomly labeling 5% of the data. We then use the False Positive Rate at 95% True Positive Rate (FPR@95% TPR), a standard metric for evaluating detection accuracy. It is scaled between 0 and 1, with higher values corresponding to more mistakes (and therefore, poorer performance). We thoroughly evaluate our definition, we run experiments over varying conditions including different regularization parameters.

Figure 2 shows a subset of our results (with the full results in Figure 7a and Figure 8). Our results highlight two key findings. **First**, the original memorization definition results in consistently high FPR, often exceeding 25%, across all datasets. This indicates that the definition frequently fails to distinguish random-label points (which are truly memorized) from natural training data points (which may generalize). Such misclassifications casts doubt on the marginal utility findings of Feldman & Zhang (2020), which employs on the original definition to identify memorized examples.

**Second**, our modified definition significantly reduces the FPR across all settings. This suggests that our method makes fewer mistakes when identifying truly memorized points. To further assess its robustness, we evaluate both definitions under varying conditions. These include evaluating over different regularization pa-

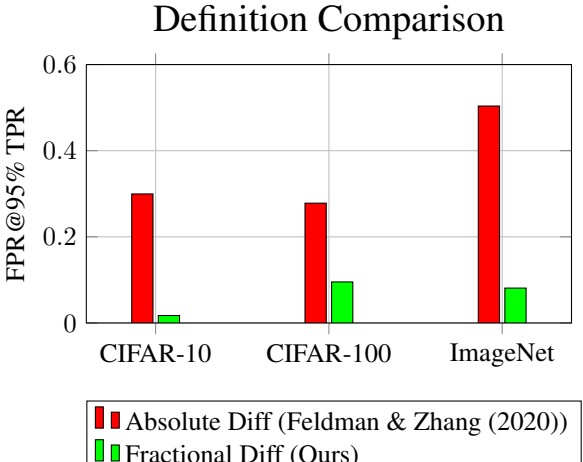

Figure 2: The original definition proposed by Feldman & Zhang (2020) has a high FPR (i.e., it makes many mistakes). However, we can reduce this significantly by using fractional difference as shown in Equation 2.

rameters (Figure 8) and comparing AUCs across the entire ROC curve (Figure 7a) and Precision-Recall curves (Figure 7b). We find that our modified definition consistently outperforms the original. This reinforces our claim that the revised scoring function better captures the true nature of memorization.

Proposed Solution: These findings underscore that the original definition is unreliable, due to its high FPR. As a consequence, it is imperative to use a low-FPR scoring function (specifically the one we propose) when estimating the marginal utility of memorized points. Without a reliable definition, non-memorized points may be mistakenly grouped with memorized ones, leading to flawed conclusions about the relationship between memorization and generalization.

---

[1] We considered two alternative denominators: either $\Pr_{h\leftarrow A(S)}[h(x_i) = y_i]$ or $\Pr_{h\leftarrow A(S\setminus i)}[h(x_i) = y_i]$. However, using the sum of the two terms is known to provide greater numerical stability Wu & Baleanu (2018), and was therefore selected as our final choice.

### 3.3 Unprincipled Memorization Threshold Leads to Inconsistent Results

Once the memorization scores have been calculated, Step 4 of the approximation algorithm (Section 2.2) is used to decide how high the memorization scores need to be to mark the points as memorized? Unfortunately Feldman & Zhang (2020) do not provide any threshold on how to differentiate memorized points from non-memorized ones.

This is an issue, as their findings, and that of their earlier theoretical work Feldman (2020), are based on the *premise* that Equation 1 provides a precise representation of memorized points (Section 2.1). We argue that the key limitation is not in their derived proofs, but in absence of a statistically robust threshold as the cornerstone of their work. For example, points with high memorization scores can behave in contradictory ways to ones with low memorization scores. Specifically, the authors show that removing memorized points with high scores (i.e., using a high memorization threshold) does not impact test accuracy. However, removing memorized points with low scores does impact test accuracy. Therefore, conclusions derived from one threshold can easily be disproved with points from a different threshold. A natural question arises: which one of the two thresholds really represents the memorized data? Do we base our conclusion on the high thresholds and claim that memorization does not impact accuracy? Or based it on the low thresholds and argue the opposite?

We answer this question in a statistically robust manner. Influenced from earlier works Arpit et al. (2017); Carlini et al. (2019); Chatterjee (2020; 2018), we propose using a *null distribution* of memorization scores derived from randomly labeled training points. Specifically, let $\mathcal{S}_{\text{rand}}$ denote the null distribution or the set of memorization scores computed for randomly labeled data points, and let $s(x)$ be the memorization score of a natural (i.e., non-random) data point $x$. We define a threshold $\tau$ based on the $(1 - \alpha)$-quantile of the null distribution:

$$\tau = \text{Quantile}_{1-\alpha}(\mathcal{S}_{\text{rand}}) \tag{3}$$

We then classify a data point $x$ as memorized if $s(x) > \tau$. For example, setting $\alpha = 0.05$ corresponds to selecting the 95th percentile of $\mathcal{S}_{\text{rand}}$ as the threshold. This ensures that only those points with memorization scores significantly higher than what would be expected under random labeling are identified as memorized. This method provides a principled and interpretable statistical criterion for distinguishing memorization from generalization.

Figure 3 provides an illustration of our proposed method for identifying memorized points using a null distribution. In this conceptual example, the blue curve represents the distribution of memorization scores for randomly labeled training points, while the orange curve shows scores for natural points. The red dashed line denotes the 95th percentile of the null distribution, which serves as the threshold for identifying memorization. Any point with a score exceeding this threshold is classified as memorized with high statistical confidence. While the distributions in this figure are synthetic, the example demonstrates how our method cleanly separates memorized points from those that generalize, and highlights the importance of grounding memorization definitions in statistically meaningful baselines.

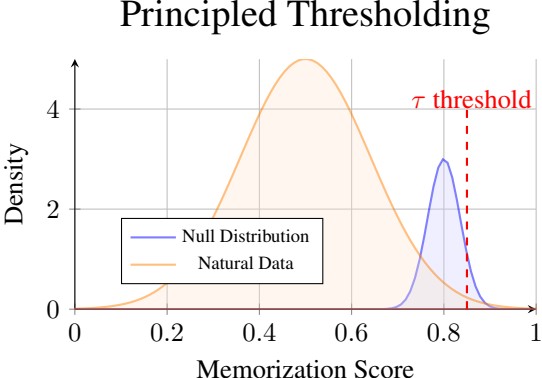

Figure 3: Illustration of our proposed null-based memorization threshold.

Proposed Solution: In the next section, when we evaluate the relationship between and generalization, we employ this statistical threshold to identify memorization. Additionally, we run further experiments using decreasing thresholds (like the original work) for completeness.

### 3.4 Data Leakage leads to overestimation of test accuracy

Once the points are identified, Feldman & Zhang (2020) now calculate the marginal utility of the memorized points by quantifying test set accuracy in the final step (Section 2.2). However, here too, we find the presence of a common oversight from the ML literature, known as data leakage. This is when there is overlap between training and test data, such as the presence of duplicate or near-duplicate example and is a well-known threat to the validity of generalization estimates Kaufman et al. (2012). It has been documented in several popular datasets used in the literature, including CIFAR and ImageNet Recht et al. (2018). Data leakage artificially inflates test accuracy by allowing models to "peek" at the test set during training, creating the illusion of strong generalization. However, this is misleading, as true generalization is defined by performance on *unseen* data.

To assess whether data leakage plays a role in the memorization study by Feldman & Zhang (2020), we conducted a preliminary investigation of the CIFAR-100 dataset using influence scores from Feldman & Zhang (2020). We found numerous cases where training points with high memorization scores had near-duplicate counterparts in the test set, shown in Figure 4. In total, CIFAR-10 and 100 contain the presence of 8% Recht et al. (2018) and 10% Barz & Denzler (2020) duplicates.

This observation raises a crucial point: removing memorized points may reduce test accuracy not because memorization is essential for generalization, but because those points directly overlap with duplicates in the test set. In other words, the accuracy drop may reflect the removal of train-test duplicates, rather than a loss of generalization ability. This issue is further exacerbated by the fact that such duplicates constitute a substantial fraction of the training data, meaning that their removal disproportionately eliminates samples that the model is evaluated on verbatim, thereby artificially depressing test accuracy.

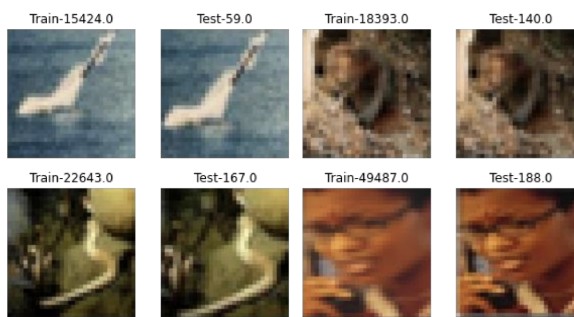

Figure 4: Data leakage: training points with high memorization scores that have near-duplicate counterparts in the test set.

Proposed Solution: While this experiment does not fully resolve the relationship between memorization and generalization (which we explore further in the next section), it highlights the importance of eliminating data leakage when interpreting results.

## 4 Re-Evaluating Marginal Utility

Having identified the different sources of error, we now implement the necessary fixes, identify the correct memorized points, and observe their impact on model accuracy.

### 4.1 Setup:

In order to perform the fairest evaluation, we re-run the experiments from the original paper and employ a similar experimental setup to the original paper (Details in Appendix A). To avoid sampling bias, we use a very high sampling rate (0.95) in order to minimize the approximation error. We use fractional difference (Equation 2) to avoid false positives when identifying memorization scores. We use our principled thresholding method to identify the memorized points which amounted to 5%, 13% and 36% of of the total training data for CIFAR-10, CIFAR-100, and ImageNet. Finally, to avoid data leakage, we use the test datasets that were purposefully designed to avoid train-test duplicates from previous works Barz & Denzler (2020); Recht et al. (2019). We provide a outline of their methodology in Appendix A.2.

To evaluate whether memorization is necessary for generalization, we train models under each of the two conditions, identical to the original Feldman & Zhang (2020) work: 1) Memorized-point removal: all identified memorized points were removed from the training set. 2) Random-point removal: an equal number of training points were removed at random, serving as a control. This

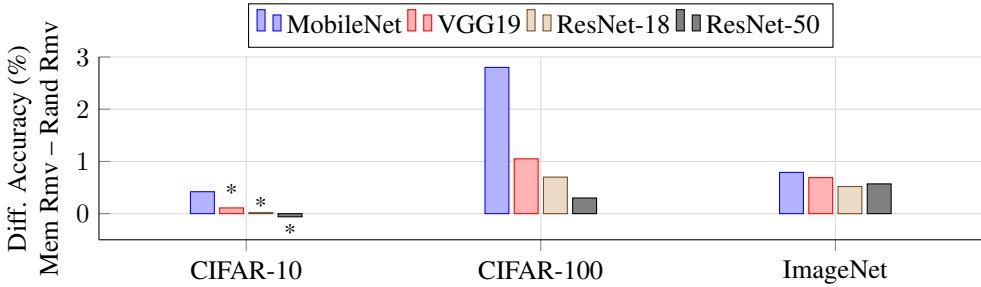

Figure 5: Change in accuracy when removing memorized points vs an equal number of random ones. * indicates that the change is not statistically significant (via a two-sided t-test), meaning the distributions are identical. Our results show the impact of removing memorized points: performance is unchanged at worst (CIFAR-10 on VGG19 and ResNet) and significantly improved at best (CIFAR-100, ImageNet).

ensures that any observed differences in performance are attributable to the specific removal of memorized points rather than simply training with fewer data. We report the difference in test accuracy between the two conditions (test accuracy when memorized points removed vs. test accuracy when random points removed). To assess statistical reliability, we conducted paired t-tests across multiple training runs for each dataset, reporting the two-sided p-values. We conduct our experiments over four different models to validate our results.

## 4.2 RESULTS:

Our results can be seen in Figure 5. As a reminder, we report the difference in test accuracy between removing memorized points and removing random points, following the original work. Positive values indicate that removing memorized points improves test accuracy (and vice-versa). We provide exact scores and $p$ values for the associated $t$-test in the Appendix (Table 3). There three major takeaways from out results:

- **In most cases, removing memorized points improves accuracy:** In most cases (11 out of 12), removing memorized points yields a statistically significant increase in accuracy. This effect is observed across all models trained on ImageNet and CIFAR-100, and for MobileNet on CIFAR-10. For example, CIFAR-100 and ImageNet, we can observe an average increase of 1.2%. In each of these cases, the improvement is statistically significant, with $t$-tests producing $p$-values of less than 0.05 (Table 3). This means that removing memorized points improves in accuracy in most settings.

- **In the remaining cases, removing memorized points has no effect:** In 3 of the 12 cases (entirely in CIFAR-10), we observe that removing memorizing points leads to a small statically-insignificant change in accuracy, indicated by * in Figure 5. The $t$-tests produce $p$-values are greater than 0.05 (Table 3). This means that though the differences might either positive or negative, they are not statistically significant and can be considered equivalent to zero.

- **The more complex the dataset, the greater the difference:** We observe that the impact of removing memorized points becomes more pronounced as dataset complexity increases. For example, ImageNet and CIFAR-100 shows significant improvements in accuracy across all models. In contrast, CIFAR-10 shows the smallest effect. This indicates that the presence of memorized points can, in fact, reduce accuracy. This suggests that in larger and more complex datasets, memorized points may play a substantial role in degrading performance.

## 5 ABLATION STUDY

Having shown that removing memorized points improves model accuracy, we now perform an ablation study to see which one of the four factors plays the largest role.

Table 1: Score and $\Delta$ for all methodological variants across datasets.

| | CIFAR-10 | | CIFAR-100 | | ImageNet | |
| Setting | Score | $\Delta$ | Score | $\Delta$ | Score | $\Delta$ |
|---|---|---|---|---|---|---|
| Baseline (Feldman and Zhang setup) | -1.71 | - | -1.25 | - | -3.99 | - |
| + Fractional Difference | -1.31 | 0.4 | -1.17 | 0.09 | -5.02 | -1.0 |
| + DeDuplicated Test | -2.24 | -0.53 | -0.91 | 0.35 | 0.38 | 4.59 |
| + 0.95 Sampling Rate | -1.15 | 0.54 | -0.44 | 0.82 | -1.68 | 3.58 |
| + Principled Thresholding | 0.12 | 1.82 | -0.04 | 1.22 | -4.03 | 1.16 |
| **All Components Changed** | **-0.1057** | **1.59** | **0.3** | **1.56** | **-0.1057** | **3.84** |

## 5.1 SETUP

We employ the experimental setup described in Section 4. Specifically, we identify memorized points on CIFAR-100 using ResNet-50 models trained with the same criteria as in Feldman et al. We use a sampling rate of 0.7 and mark points with a memorization score above 0.25 (under the original definition) as memorized. We then remove these memorized points from the training set, retrain the model, and evaluate test accuracy on the original test set (i.e., without de-duplication). This configuration serves as our baseline. Next, we introduce one improvement at a time to each of the four factors considered in our work (sampling bias, fractional-difference definition, principled thresholding, and data leakage) and, for each modified configuration, retrain the model and measure the resulting change in utility (over 50 models) and improvement over the baseline. Finally, we run the two sided t-test to see which factors introduce a statically significant improvement in utility over the baseline.

## 5.2 RESULTS

Our results are shown in Table 1. We can make a few observations: **Combining all four components provides the strongest results.** The "All Components Changed" row shows that the combined system almost always outperforms baseline across all datasets, with deltas around +1.5 to +3.8. This implies the corrections are additive rather than redundant, meaning each modification addresses a different failure mode in the original Feldman setup. **The four factors do not contribute equally.** Principled thresholding and improved sampling rate provide the largest improvement over the baseline. In stark contrast, the fractional-difference definition and de-duplicating the test set introduces only modest improvements that are statistically insignificant. **Thresholding appears to be a "bottleneck fix."** Principled thresholding alone provides consistent improvement across all datasets. This indicates that the use of the arbitrary thresholding step in Feldman & Zhang (2020), where they choose the 0.25 threshold for memorization without any justification, is the primary source of the observed drop in utility. This is likely because unprincipled thresholding can cause non-memorized points to be misclassified as memorized, thereby removing samples that actually support generalization.

**Overall, our ablation shows that the failure mode of the original Feldman & Zhang (2020) method is not due to any single component, but rather to the cumulative effect of several small sources of error.** Individually, each fix only partially reduces the negative utility, but together they correct the underlying instability and reverse the sign of the effect. Once these sources of error are corrected, we find that removing memorized points improves generalization.

## 6 DISCUSSION

**Correcting Misconceptions from Prior Work:** Taken together, our results show that removing memorized points never harms performance (CIFAR-10) and can also significantly improve accuracy on more complex datasets (CIFAR-100, ImageNet). This directly contrasts with the original work Feldman & Zhang (2020), which reported that removing memorized points reduced accuracy. The discrepancy arises because their analysis suffered from four methodological flaws (Section 3): sampling bias, a high false positive rate, unprincipled thresholding, and data leakage. Correcting

for these issues reveals that memorized points are **not** necessary for generalization; on the contrary, their removal can improve test performance.

**Commonly Overlooked Errors:** The four errors we identify in our work sampling bias, flawed definitions, unprincipled thresholding, and data leakage, are not unique to Feldman & Zhang (2020), but can be considered common oversights in machine learning. Each issue stems from methodological shortcuts that, while convenient, can seriously distort the final conclusions. Our findings highlight that careful choices of sampling rates, principled scoring functions, statistically grounded thresholds, and rigorous checks for leakage are essential. Without such rigor, small methodological lapses risk compounding into broad conceptual errors in our understanding.

**Impact of Dataset Complexity:** Our results show that memorization becomes increasingly harmful as training data complexity grows. We observed the largest accuracy gains from removing memorized points on ImageNet, followed by CIFAR-100, and the smallest gains on CIFAR-10. Extending this trend to modern large scale datasets containing billions of examples (e.g., LAION Schuhmann et al. (2022)), we hypothesize that the negative impact of memorization is likely to be even stronger. This provides a compelling motivation for curating datasets to identify and filter memorized points. Otherwise, practitioners risk leaving significant performance improvements unrealized.

**Memorization and Privacy:** Memorization has a direct implication for security and privacy research. This is because one hand, Feldman (2020) claim that memorization was needed for generalization while other works demonstrated that memorization leads to lead to leakage of private data, model stealing attacks, increased model bias, poor robustness to distribution shifts, and susceptibility to adversarial samples. In other words, generalization and security/privacy can not be simultaneously achieved. While this might have dissuaded some researchers, our work shows that this tension does not exist. This is because memorization is not necessary for generalization. Future researchers are encouraged to explore methods to build models that both generalize and are private.

## 7 RELATED WORK

A seminal work on memorization in deep learning was by Zhang et al. (2017), who demonstrated that neural networks can fit unstructured data, including random Gaussian noise. This finding sparked a long-standing tension between memorization and generalization. Early studies argued for limiting memorization to encourage genuine pattern learning, partly because memorization introduces privacy risks such as membership inference (Carlini et al., 2019; 2022). In response, several strategies were proposed: regularization (Arpit et al., 2017), filtering weak gradients (Zielinski et al., 2020; Chatterjee, 2020), and adjusting model size (Arpit et al., 2017; Zhang et al., 2019). While effective at reducing memorization, these methods often came at the cost of model accuracy.

Despite these efforts, the true impact of memorization on model behavior remained unclear. Addressing this first required methods to identify memorized points. Several post hoc approaches were developed, including clustering based techniques (Stephenson et al., 2021), repurposed membership inference attacks (Carlini et al., 2022), and pseudo LOO methods (Feldman & Zhang, 2020). Equipped with these tools, researchers could begin studying the role of memorization in model efficacy. While some recent work suggests that memorization, when coupled with spurious correlations, can harm generalization (Bayat et al., 2024), the most influential perspective came from Feldman & Zhang (2020), who argued that memorization was, in fact, necessary for generalization. This conclusion shaped much of the subsequent discourse but was partly an artifact of methodological issues. By correcting these errors and rerunning the experiments, our results instead show the opposite: memorization either harms generalization or has no impact at all.

## 8 CONCLUSION

Memorization is the ability of the model to fit labels to seemingly random samples. Recent work from Feldman & Zhang (2020) demonstrated that memorization is necessary for generalization. We show that the original work suffered from a number of crucial errors. These include sampling bias, high false positives rate, unprincipled thresholding, and data leakage. Having accounted for these errors with proper fixes, we find that removing memorized points either improves accuracy or has no impact at all. This in stark contrast to the original work, who reached the incorrect conclusion

due to the errors we identify in this work. Looking forward, the challenge is not to embrace memorization as an unavoidable feature of modern models. Instead, it is to design architectures, training procedures, and evaluation protocols that minimize its impact.

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

# A APPENDIX

## A.1 TRAINING SETUP

Our training setup exists of using 100 epochs (CIFAR-10 and 100) and 75 epochs (ImageNet), using a batch size of 512, with a triangular learning rate of 0.4. We train an average of 275 models for sampling bias experiments (Figure 1) and definition comparison experiments (Figure 2). We find this is sufficient to produce consistent results. When comparing definitions, we select points that are absent from at least approximately 10 models. This is done to ensure that a point that is absent from just a single model does not bias the resulting score calculations. However, we train substantially more models when the sampling rate is 0.95. Approximately 500 for CIFAR-10/100 and at least 150 for ImageNet. We found that training these numbers of models was sufficient since memorization scores tend to stabilize much earlier (Figure( 9). To calculate marginal utility, we train 100 models for Cifar-10/100 and 50 for ImageNet.

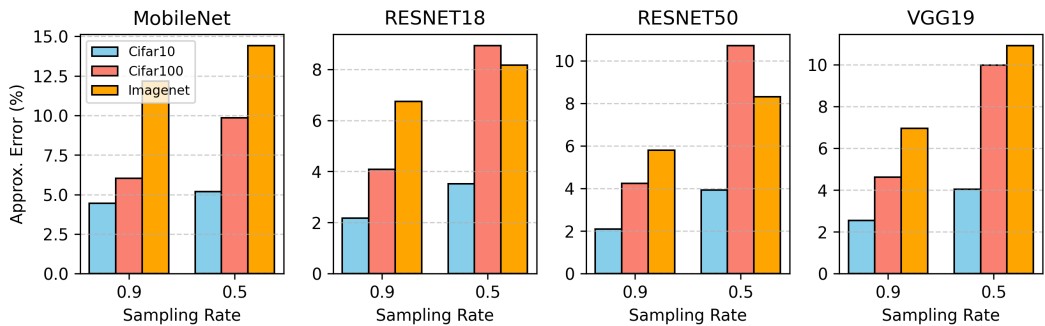

Figure 6: Sampling bias: Higher sampling rates result in more accurate scores. While lower rates lead to higher approximation error.

## A.2 DATA LEAKAGE REMOVAL

We did not perform our own near-duplicate detection; instead, we relied on the deduplicated test sets released by Barz & Denzler (2020) and Recht et al. (2018), both of whom identify duplicates using cosine similarity in deep feature space rather than pixel-level distance. Barz & Denzler (2020) report up to 3% duplicates in CIFAR-10 and up to 10% in CIFAR-100, while Recht et al. (2018) later found that CIFAR-10 contains roughly 8% duplicates, substantially more than previously reported, and manually curated fully deduplicated test sets for CIFAR-10 and ImageNet to eliminate this issue. Because no curated CIFAR-100 set was released by Recht et al. (2018), we adopt the deduplicated version from Barz & Denzler (2020) and do not recompute similarity thresholds, per-class removal counts, or pixel-space statistics, since doing so would introduce a different notion of duplication and risk class-specific bias.

Table 2: Symbols used and their meanings.

| Symbol | Meaning |
|---|---|
| $x_i$ | training data point |
| $y_i$ | training point label |
| $x'_i$ | test data point |
| $y'_i$ | test point label |
| $S$ | training set |
| $A$ | training algorithm |
| $n$ | size of the training set |
| $m$ | number of points removed from the training set |
| $h$ | trained model |
| $t$ | trial |

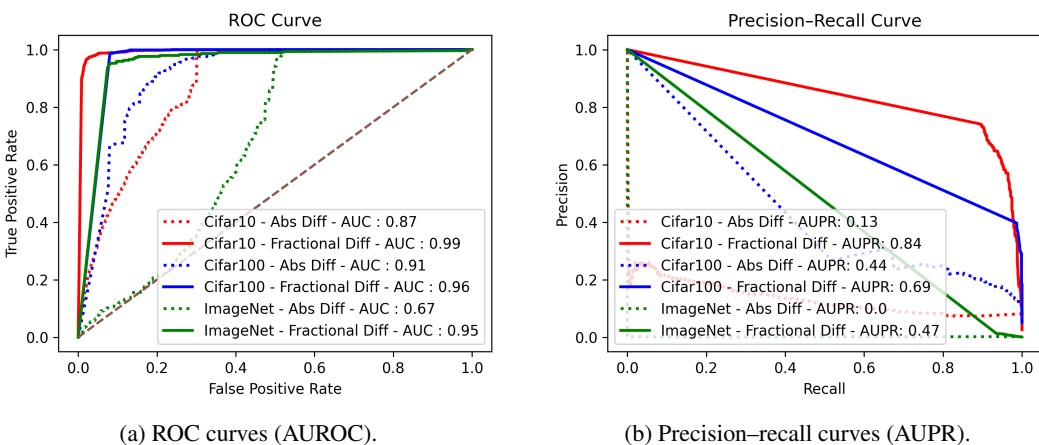

(a) ROC curves (AUROC).

(b) Precision–recall curves (AUPR).

Figure 7: Memorization definitions evaluated via ROC and precision–recall curves on CIFAR-10, CIFAR-100, and ImageNet. Our fractional-difference definition consistently outperforms the original score, achieving higher AUROC and AUPR across all datasets.

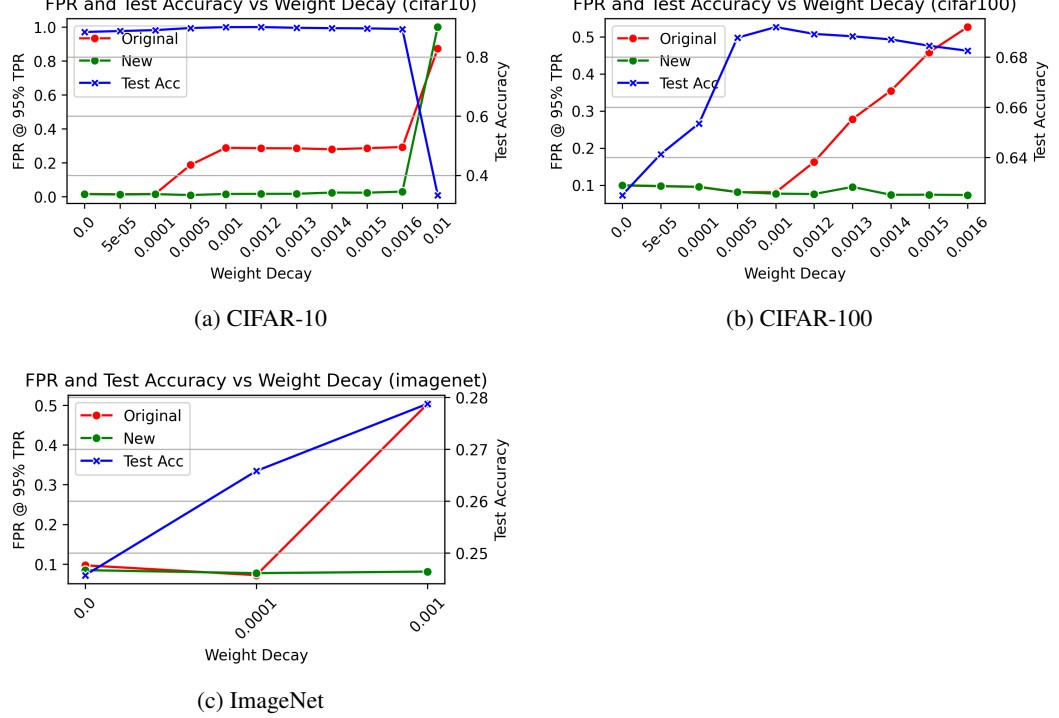

(a) CIFAR-10

(b) CIFAR-100

(c) ImageNet

Figure 8: Visual comparison of results across datasets. The final quadrant is left intentionally blank.

Table 3: Comparison of model accuracy (in %) after removing memorized samples versus randomly selected samples, evaluated across datasets (CIFAR-10, CIFAR-100, ImageNet) and architectures (MobileNet, VGG19, ResNet18, ResNet50).

| Dataset | Model | Mem Remove Acc (%) | Rand Remove Acc (%) | Difference ($\pm$ std) | p-value | Significance |
|---------|-------|--------------------|---------------------|------------------------|---------|--------------|
| **CIFAR-10** | | | | | | |
| | MobileNet | 69.32 | 69.01 | $0.3 \pm 1.6$ | $2.67 \times 10^{-2}$ | Significant |
| | VGG19 | 82.31 | 82.04 | $0.26 \pm 2.46$ | $3.5 \times 10^{-1}$ | Not Significant |
| | ResNet18 | 83.86 | 83.80 | $0.06 \pm 0.67$ | $1.1 \times 10^{-1}$ | Not Significant |
| | ResNet50 | 84.82 | 84.97 | $-0.07 \pm 0.78$ | $9.2 \times 10^{-1}$ | Not Significant |
| **CIFAR-100** | | | | | | |
| | MobileNet | 44.05 | 41.13 | $2.9 \pm 1.6$ | $1.2 \times 10^{-42}$ | Significant |
| | VGG19 | 60.45 | 59.39 | $1.05 \pm 0.51$ | $1.4 \times 10^{-42}$ | Significant |
| | ResNet18 | 64.88 | 64.16 | $0.71 \pm 0.45$ | $5.12 \times 10^{-39}$ | Significant |
| | ResNet50 | 67.75 | 67.47 | $0.27 \pm 1.23$ | $1.4 \times 10^{-2}$ | Significant |
| **ImageNet** | | | | | | |
| | MobileNet | 7.29 | 6.5 | $0.79 \pm 0.24$ | $1.7 \times 10^{-43}$ | Significant |
| | VGG19 | 10.63 | 9.94 | $0.69 \pm 0.19$ | $7.07 \times 10^{-43}$ | Significant |
| | ResNet18 | 22.70 | 21.40 | $1.3 \pm 0.44$ | $1.35 \times 10^{-37}$ | Significant |
| | ResNet50 | 23.84 | 22.74 | $1.1 \pm 1.96$ | $1.07 \times 10^{-4}$ | Significant |

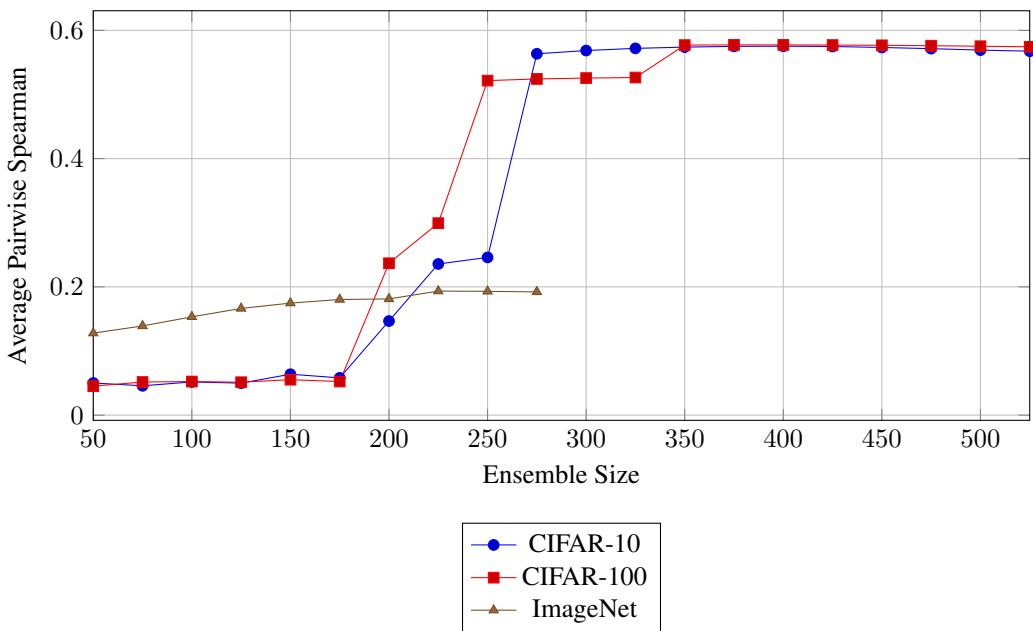

Figure 9: Average pairwise Spearman correlation between memorization scores as a function of ensemble size. The figure indicates that memorization scores stabilizes around 275, 350, and 225 models for CIFAR-10, 100, and Imagenet.

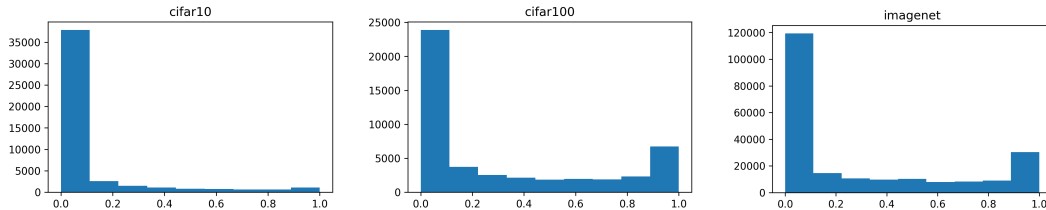

Figure 10: Distribution of memorization scores after removing the identified memorized points. We can see that the data distributions still show the long tailed structure, which is a necessary assumption for Feldmen et al's works.

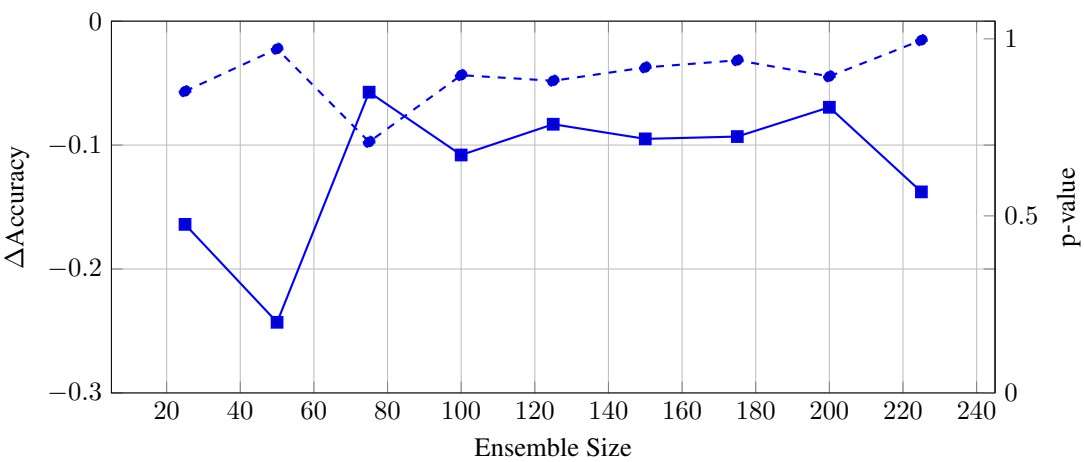

(a) CIFAR-10: The plot shows how the difference in accuracy and the corresponding p-values vary with ensemble size. We can observe that p-value remains constantantly high across model ensables indicating that removing memorized points does not impact model accuracy.

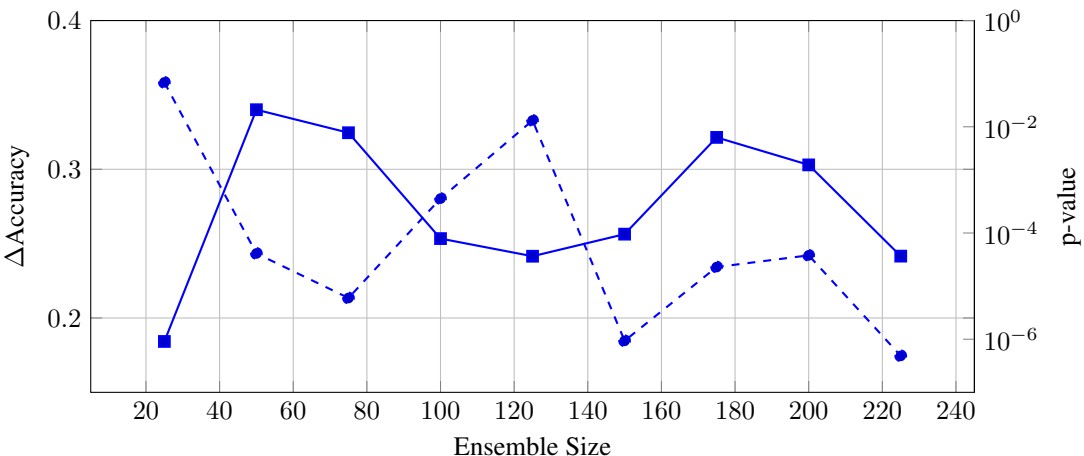

(b) CIFAR-100: The plot shows how the difference in accuracy and the corresponding p-values vary with ensemble size. We can observe that p-value remains constantantly low across model ensables indicating that removing memorized points does improves model accuracy.

Figure 11: Ensemble size vs Utility

Table 4: Effect of methodological changes on memorization measurements across datasets.

| Dataset | Setting | Score | $\Delta$ | p-value |
|---------|---------|-------|----------|---------|
| 6*CIFAR-10 | Original Feldmen et al (Baseline) | -1.7162 | 0.0 | 1.0 |
| | + Fractional Difference Definition | -1.3106 | 0.4 | 1.25e-06 |
| | + DeDuplicated Test Set | -2.2425 | -0.53 | 8.04e-05 |
| | + 0.95 Sampling Rate | -1.1538 | 0.54 | 2.00e-09 |
| | + Principled Thresholding | 0.1227 | 1.82 | 6.14e-23 |
| | All four components changed | -0.1057 | 1.59 | 8.92e-86 |
| 6*CIFAR-100 | Original Feldmen et al (Baseline) | -1.2574 | 0.0 | 0.0 |
| | + Fractional Difference Definition | -1.1762 | 0.09 | 0.6064 |
| | + DeDuplicated Test Set | -0.9194 | 0.35 | 0.0971 |
| | + 0.95 Sampling Rate | -0.4462 | 0.82 | $9.21 \times 10^{-4}$ |
| | + Principled Thresholding | -0.0419 | 1.22 | $5.54 \times 10^{-9}$ |
| | All four components changed | 0.3 | 1.56 | $1.03 \times 10^{-2}$ |
| 6*ImageNet | Original Feldmen et al (Baseline) | -3.995 | 0.0 | 1.0 |
| | + Fractional Difference Definition | -5.0273 | -1.0 | 1.20e-01 |
| | + DeDuplicated Test Set | 0.3837 | 4.59 | 3.12e-08 |
| | + 0.95 Sampling Rate | -1.6823 | 3.58 | 5.63e-02 |
| | + Principled Thresholding | -4.0379 | 1.16 | 9.84e-01 |
| | All four components changed | 0.54 | 4.53 | 1.10e-32 |

