# OpenReview forum: "Is Memorization Actually Necessary for Generalization"
_ICLR.cc/2026/Conference — Submitted to ICLR 2026_

### Official Review · Reviewer_XWnb · 2025-10-28

**Soundness:** 2
**Presentation:** 3
**Contribution:** 3
**Rating:** 4
**Confidence:** 4

**Summary:**

The paper revisits and challenges the claim by Feldman & Zhang (2020) that memorization is essential for neural network generalization. It  argues that the original conclusions were based on four critical methodological flaws: (1) sampling bias from using a low sampling rate that inflated memorization scores, (2) a high false positive rate due to an imprecise memorization definition, (3) unprincipled thresholding that lacked a statistically grounded criterion for distinguishing memorized from non-memorized samples, and (4) data leakage between training and test sets that artificially increased measured accuracy. To address these issues, the paper introduces methodological corrections, including higher sampling rates approaching leave-one-out, a new fractional difference definition to reduce false positives, a null-distribution-based threshold for principled identification of memorized points, and leakage-free test datasets. Through experiments on CIFAR-10, CIFAR-100, and ImageNet across multiple architectures (MobileNet, VGG19, ResNet18, and ResNet50), the paper shows that removing truly memorized samples never harms and often improves test accuracy, particularly for more complex datasets. These results overturn the notion that memorization is necessary for generalization, suggesting instead that it can degrade performance, and the paper advocates viewing memorization as a behavior to mitigate rather than a property to preserve in deep learning models.

**Strengths:**

1.  The paper tackles an influential and widely cited claim about the necessity of memorization for generalization, offering a fresh and critical perspective.

2. It systematically identifies four core methodological flaws—sampling bias, high false positive rate, unprincipled thresholding, and data leakage—in the original Feldman & Zhang (2020) study.

3. The paper proposes concrete fixes, including higher sampling rates, a fractional-difference definition of memorization, statistically grounded thresholding via null distributions, and de-duplicated test datasets.

4. The experimental evaluation spans multiple datasets (CIFAR-10, CIFAR-100, ImageNet) and architectures (MobileNet, VGG19, ResNet18, ResNet50), providing empirical evidence.

5. The paper clearly articulates the logical flow from identifying issues to proposing corrections and presenting re-evaluated results.

**Weaknesses:**

1. In section 3.2, the paper treats randomly relabeled samples as the ground truth for memorized points when comparing definitions. While convenient, this synthetic setup may not reflect real-world memorization behaviors (e.g., rare or atypical but semantically consistent samples), limiting the validity of the conclusions about false positive rates.

2. The paper does not release code for reproducing key experiments, limiting transparency and reproducibility.

3. In Section 3.4, the paper identifies train–test overlaps as a potential source of inflated accuracy and supports this claim with qualitative examples (Figure 4). However, it does not quantify the extent of this leakage — e.g., how many images overlap in CIFAR or ImageNet, or what proportion of the test set is affected. Without a concrete measurement, it is difficult to assess whether data leakage is a widespread issue or limited to a few instances, weakening the strength of the argument.

4. Figure 6 shows that the approximation error for CIFAR-100 is higher than for CIFAR-10, even though the two datasets are structurally similar and differ mainly in the number of classes. The paper does not explain why this happnes. A deeper analysis of how dataset complexity affects sampling bias would strengthen the paper’s claims.

5. In Figure 6, the approximation error for CIFAR-10 appears nearly unchanged between sampling rates of 0.5 and 0.9, suggesting that increasing the sampling rate does not substantially reduce the error for this dataset. The paper does not explain why this occurs. Additionally, the results for ImageNet are not reported, leaving it uncertain whether similar trends hold for larger and more complex datasets.

6. While Feldman & Zhang (2020) trained on the order of thousands of models to approximate the memorization score, this paper uses 50 models (section 4.1). Although computationally more feasible, this smaller number of models may not fully capture the variability needed for precise memorization-score estimation.

Minor weaknesses:
1. In section 3.4 there are some "?" that indicate some references might not be cited properly.

**Questions:**

1. Ground truth for memorization: In Section 3.2, you use randomly relabeled samples as a proxy for “truly memorized” points. Could you elaborate on why this choice is representative of real-world memorization? Have you considered alternative or complementary definitions, such as using rare or low-density samples as ground-truth memorized points?

2. Extent of data leakage: In Section 3.4, Figure 4 presents qualitative examples of train–test overlaps, but the paper does not quantify the extent of this issue. Could you provide statistics on how many overlapping or near-duplicate samples were detected in CIFAR-10, CIFAR-100, and ImageNet? Knowing whether the leakage affects a small subset or a substantial portion of the test set would greatly strengthen your argument.

3. Sampling rate behavior across datasets: Figure 6 shows that CIFAR-100 exhibits higher approximation error than CIFAR-10, even though the datasets are structurally similar. Could you clarify what drives this discrepancy?

4. Unclear trend for CIFAR-10 and missing ImageNet results: In Figure 6, the approximation error for CIFAR-10 appears almost constant between sampling rates of 0.5 and 0.9, which seems inconsistent with the claim that higher sampling rates reduce bias. Can you explain why this occurs? Also, could you include or discuss corresponding results for ImageNet to assess whether this trend generalizes to larger datasets?

5. Model number: In Section 4.1, you mention training approximately 50 models. Could you clarify how you determined that this number was sufficient for stable memorization estimates? Have you evaluated whether increasing the number of models meaningfully changes the observed trends or variance of the results?

6. Reproducibility: Since no code is released, would you consider sharing implementation details, hyperparameters, and scripts? This would help readers verify the findings and reproduce key experiments.

---

> ### Author Response · Authors · 2025-11-23
>
> **"Ground truth for memorization: In Section 3.2, you use randomly relabeled samples as a proxy for “truly memorized” points. Could you elaborate on why this choice is representative of real-world memorization? Have you considered alternative or complementary definitions, such as using rare or low-density samples as ground-truth memorized points?"**
>
> Thank you for the question. We use randomly labeled samples as a proxy for ground-truth memorization because (1) they are a common means of simulating memorization in current literature [1,2,3,4,5,6] (2) they provide a controlled and model-agnostic way to induce true memorization: the only way a model can achieve low training loss on these points is by memorizing their input–label pairs, since the labels contain no recoverable structure. This makes them a reliable diagnostic tool for evaluating whether a memorization score correctly identifies points that are learned purely by rote.
> Alternative choices—such as using visually rare or low-density samples—do not provide a clean ground truth because rarity does not necessarily imply memorization, and models can often learn such samples without overfitting them. In contrast, random labels guarantee the absence of any meaningful label–feature relationship, which is precisely the condition under which memorization becomes the only training mechanism.
>
> That said, our goal is not to claim that random labels are the only or exclusive notion of memorization. Rather, they serve as a complementary stress test that cleanly isolates memorization from other phenomena such as sample difficulty or class imbalance. We agree that examining low-density or rare samples is interesting in its own right, but these definitions capture a different notion, visual atypicality, rather than true memorization. Our method specifically aims to detect model–label instability, and random relabeling provides the clearest ground-truth signal for that purpose.
>
> [1] Zhang et al. 2017
>
> [2] Arpit et al. 2017
>
> [3] Stephenson et al. 2021
>
> [4] Carlini et al. 2019
>
> [5] Zielinski et al. 2020
>
> [6] Chatterjee, 2020
>
> **"Extent of data leakage: In Section 3.4, Figure 4 presents qualitative examples of train–test overlaps, but the paper does not quantify the extent of this issue. Could you provide statistics on how many overlapping or near-duplicate samples were detected in CIFAR-10, CIFAR-100, and ImageNet? Knowing whether the leakage affects a small subset or a substantial portion of the test set would greatly strengthen your argument."**
>
> We did not run our a duplicate detection procedure ourselves. Instead, we relied entirely on prior, well-established deduplication efforts by Barz & Denzler (2020) and Recht et al. (2019). Both works identify near-duplicate test images using l2 distance in deep feature space, not pixel-level distance. Barz et al. identifyed 3% and 10% duplicates in CIFAR-10 and CIFAR-100, while Recht et al. found an additional 5% duplicates in CIFAR-10, that were missed by Barz et al, bringing the total to duplicates to 8% of CIFAR-10 dataset. We have now added this information to the manuscript.
>
> **"Sampling rate behavior across datasets: Figure 6 shows that CIFAR-100 exhibits higher approximation error than CIFAR-10, even though the datasets are structurally similar. Could you clarify what drives this discrepancy?"**
>
> Thank you for highlighting this observation. The key driver of the discrepancy is the difference in generalization gap between CIFAR-10 and CIFAR-100. Models trained on CIFAR-100 exhibit a substantially larger train–test gap than those trained on CIFAR-10. A larger gap implies that the model relies more heavily on memorization to fit the training distribution [1]. Because our approximation error effectively measures how sensitive the memorization score is to subsampling, a model that memorizes more will naturally show a steeper degradation under reduced sampling rates.
>
>
> [1] Examining Natural Memorization in Deep Learning Models

---

> ### Author Response · Authors · 2025-11-23
>
> **"Unclear trend for CIFAR-10 and missing ImageNet results: In Figure 6, the approximation error for CIFAR-10 appears almost constant between sampling rates of 0.5 and 0.9, which seems inconsistent with the claim that higher sampling rates reduce bias. Can you explain why this occurs? Also, could you include or discuss corresponding results for ImageNet to assess whether this trend generalizes to larger datasets?"**
>
> Thank you for raising this.
> 1. Based on your feedback, we trained thousands of additional models for the sampling rate experiments, raising the average number of models per dataset from 100 to 275. And we can see the the difference in approximation error is now starker for CIFAR-10 (Figure 6). This shows that approximation error does increase with the sampling rate. However, it is simply a smaller increase compared to CIFAR-100 or ImageNet. This is expected because CIFAR-10 models exhibit relatively small generalization gaps and therefore rely less on memorization.
> 2. More complex datasets (e.g., CIFAR-100) show a much larger increase in approximation error. This is likely because models tend to memorize more data from more complex datasets. Since most real-world datasets resemble CIFAR-100 or ImageNet more than CIFAR-10 in terms of complexity, we expect sampling rate to play an even more important role outside of small benchmark settings.
> 3. Finally, we ran the ImageNet experiments in response to the reviewer’s suggestion. Figure 6 has been updated to show the results. The results confirm the same pattern: approximation error becomes substantially worse as the sampling rate decreases. This shows that the trend generalizes to larger datasets and supports our overall conclusions.
>
>
>
> **"Model number: In Section 4.1, you mention training approximately 50 models. Could you clarify how you determined that this number was sufficient for stable memorization estimates? Have you evaluated whether increasing the number of models meaningfully changes the observed trends or variance of the results?"**
> 1. Our experiments do not use only 50 models for calculating memorization scores. We had used 50 for calculating the final drop in accuracy. To calculate memorization scores, we use 500 for CIFAR-10/100 and 150 for Imagenet (Appendix A.1).
>
> 2. To ascertain whether this was enough models for stable memorization scores, we conducted a sweep over the number of models (now present in Figure-9). We find that the Spearman correlation between memorization scores stabilizes at roughly 300 models for Cifar10/100 and 150 for Imagenet. Beyond this point, the memorization scores do not change. Nevertheless, for completeness and to match the spirit of Feldman & Zhang’s large-scale setup, we trained an additional 200 models, bringing the total to 500 (for CIFAR10/100). Even though the imagenet memorization scores converge with 150 models, we are actively training more imagenet models to raise the total to 500 as well. We hope to update the paper before the end of the discussion period.
>
>
>
> **"Reproducibility: Since no code is released, would you consider sharing implementation details, hyperparameters, and scripts? This would help readers verify the findings and reproduce key experiments."**
>
> We provide the details in Appendix A.1. If the reviewer feels we are missing anything, let us know and we will be happy to provide more information.

---

### Official Review · Reviewer_AsrV · 2025-10-30

**Soundness:** 2
**Presentation:** 4
**Contribution:** 2
**Rating:** 4
**Confidence:** 5

**Summary:**

This paper challenges the conclusions of Feldman and Zhang (2020) [2] concerning the definition and
empirical role of memorization in generalization. The authors argue that several methodological factors
such as sampling bias, arbitrary threshold selection, lack of model convergence, and data leakage may have
exaggerated the reported connection between memorization and test accuracy. By introducing a cleaner
experimental pipeline together with a null distribution calibration, the paper re-examines whether
removing “memorized” samples genuinely harms model performance.

[1] Feldman, Vitaly. "Does learning require memorization? a short tale about a long tail."
[2] Feldman, Vitaly, and Chiyuan Zhang. "What neural networks memorize and why: Discovering the long
tail via influence estimation."

**Strengths:**

The paper is well written and clearly structured, making it easy to follow the motivation, methodology, and
results. The introduction of the random-label null distribution is an interesting and original idea that
provides a fresh statistical perspective on memorization. The experimental analysis is broad and
systematic, covering multiple datasets and models.

**Weaknesses:**

**1. On the meaning of “evaluated against a baseline.”**

The memorization score introduced in [2] was designed as an empirical instantiation of the theoretical
definition proposed in [1]. In that framework, the score is not a heuristic metric but a property intrinsic to
each sample, derived directly from theory. The visualizations in [2] (see https://pluskid.github.io/influence-
memorization/) clearly show that samples with high memorization scores tend to be atypical or rare from a
human perspective. Given this context, the statement on line 61 that “the memorization scoring definition is
never evaluated against a baseline” may not be conceptually meaningful, since the quantity itself serves as its
own theoretical ground truth. If the authors intend to reinterpret memorization as a detectability or
inference problem, a more suitable comparison might come from membership inference analysis (MIA),
which explicitly measures how much a model’s output depends on individual training samples. I could not
find a clear indication that the current paper engages with the MIA literature, so it would help to clarify
whether this connection was considered or deliberately excluded.

**2. On the null-distribution analysis.**

The introduction of random labels to construct a null distribution is a valuable methodological addition.
However, it would be insightful to visualize which samples receive high memorization scores under the null
distribution and whether they remain visually atypical to human observers. In addition, reporting the
correlation between memorization scores computed under the null and natural distributions would help
reveal whether the score primarily captures intrinsic sample rarity or genuine label–feature relationships.

**3. On the long-tail assumption after data cleaning.**

Both [1] and [2] derive their conclusions under the assumption that the data distribution is long-tailed
across sub-populations. Since this paper removes near-duplicates and mislabeled samples, it should verify
whether the cleaned CIFAR-10/100 datasets still exhibit such a long-tail structure. Providing the post-
cleaning memorization-score histogram or sub-population frequency-rank plot is necessary to demonstrate
that the theoretical premise of the original argument remains valid in the new experimental setting.

**4. On data-leakage detection.**

Section 3.4 briefly mentions the removal of near-duplicates but does not specify the exact detection
method. Was cosine similarity in feature space used, or pixel-level distance? Providing per-class statistics
for the number of removed examples, the similarity threshold, and total removal counts in the appendix
would improve transparency and help assess whether the cleaning process introduced class-specific bias.
Also, please check lines 330–332, where double question marks appear in the citations.

**5. On experimental comparison in Section 4.1.**

It necessary to include a direct comparison between the original [2] scoring method and the revised
method proposed here, evaluated under identical sampling rates and all other training setups. Reporting
the resulting accuracy changes after removing top-scored versus random samples for both definitions
would more clearly isolate the effect of the new metric itself.

**Questions:**

1. Could the authors clarify whether they intend to reinterpret memorization as a detectability/inference task, and if so, why membership inference analysis (MIA) was not included as a baseline comparison?

2. Can the authors report visualizations and correlation statistics comparing null-distribution scores to natural scores to assess whether the metric reflects sample rarity or label–feature alignment?

3. Could the authors provide post-cleaning memorization-score histograms or sub-population frequency-rank plots to verify that the long-tail structure still holds?

4. What similarity metric and threshold were used to remove near-duplicates, and can the authors provide per-class removal statistics?

5. Can the authors include a controlled comparison showing accuracy changes when removing top-scored versus random samples under both the original and revised scoring methods?

---

> ### Author Response · Authors · 2025-11-23
>
> **"Could the authors clarify whether they intend to reinterpret memorization as a detectability/inference task, and if so, why membership inference analysis (MIA) was not included as a baseline comparison?"**
>
> We agree that the memorization score introduced in FZ is grounded in theory and, within that framework, serves as an intrinsic property of each sample rather than a heuristic requiring a baseline for validation. Our focus is on testing whether the operational memorization score from FZ aligns with established intuitions about what constitutes memorization.
>
> In particular, randomly labeled points have historically been considered prototypical memorized examples, yet our results show that the empirical proxy from FZ fails to mark such points as memorized. This suggests limitations in the score’s ability to capture certain classes of memorization, even if it is theoretically well-defined.
>
> We acknowledge an overlap between memorization and membership inference analysis (MIA): memorized points are generally the most vulnerable to MIA attacks [1]. And that removing points that ar ememorized AND vulernable to membership inference might show the same behavioir. We will be happy to mention this in the discussion section.
>
> [1] https://openreview.net/pdf?id=EGIvMUk5duH
>
> **"Can the authors report visualizations and correlation statistics comparing null-distribution scores to natural scores to assess whether the metric reflects sample rarity or label–feature alignment?"**
>
> We appreciate this suggestion. We have added the images of the samples identified as memorized under the null distribution to the supplementary material. Notably, these high-score images appear visually typical and representative of their respective classes. They do not exhibit noise, distortions, or outlier-like patterns. This indicates that the memorization score under the null distribution is not merely detecting visually atypical or rare samples. Rather, its behavior under the natural distribution reflects instability that arises specifically from the interaction between model predictions and the true label–feature relationship. This supports our argument that the score is not functioning as a simple “rarity detector.”
>
> Regarding the request to compute the correlation between null-distribution scores and natural-distribution scores: the null distribution is used to set the cutoff threshold for flagging memorized samples, not to generate a directly comparable score scale. Because the values are not calibrated across the two distributions, a correlation would not be meaningful. If the reviewer has a particular correlation metric or comparison protocol in mind, we are happy to run that analysis.
>
> **"Could the authors provide post-cleaning memorization-score histograms or sub-population frequency-rank plots to verify that the long-tail structure still holds?"**
>
> We thank the reviewer for highlighting the importance of verifying the long‑tail assumption in our experimental setup. First, we would like to clarify that we use the deduplicated test sets provided by Barz & Denzler (2020) and Recht et al. (2019). The training set is left fully intact, so the sub‑population frequency distribution that underlies the memorization analysis in is unchanged.
>
> Second, we do not explicitly remove mislabeled training examples. Some of the points identified as “memorized” in our method may indeed carry incorrect labels, but these are not removed as part of preprocessing; they are only flagged during utility experiments.
>
> To address the reviewer’s concern directly, we now provide memorization‑score histograms (Fig. 10) computed after removing the points that our method marks as memorized. These histograms show that the long‑tail structure persists even after those removals. Specifically:
> 1. The memorization‑score distribution retains its heavy‑tailed shape.
> 2. The points we remove lie predominantly at the tip of the tail, i.e., they are the highest‑memorization‑score examples.
> 3. Importantly, the overall tail structure remains intact, indicating that the sub‑population rarity patterns emphasized by Feldman et al. are not disrupted by our cleaning.
> Together, these observations confirm that the theoretical premise of a long‑tailed distribution continues to hold in our experimental setting, even after test‑set deduplication and the identification of potentially mislabeled memorized examples.

---

> > ### Author Response · Authors · 2025-11-23
> >
> > **"What similarity metric and threshold were used to remove near-duplicates, and can the authors provide per-class removal statistics?"**
> >
> > We did not run our a duplicate detection procedure ourselves. Instead, we relied entirely on prior, well-established deduplication efforts by Barz & Denzler (2020) and Recht et al. (2019). Both works identify near-duplicate test images using L2 similarity in deep feature space, not pixel-level distance. Barz et al. provide deduplicated versions of the CIFAR-10 and CIFAR-100 test sets after identifying 3% and 10% duplicates in CIFAR-10 and CIFAR-100, while Recht et al. manually curated fully deduplicated test sets for CIFAR-10 and ImageNet after discovering substantial residual duplicates missed by automated detection. Specifically, Recht et al. found an additional 5% duplicates in CIFAR-10, that were missed by Barz et al, bringing the total to duplicates to 8% of CIFAR-10 dataset.
> >
> > Because we use the exact test sets already released by these authors, we do not reapply the feature-space matching procedure, and therefore we do not have dataset-specific similarity thresholds, pixel-distance metrics, or per-class removal statistics of our own to report. Instead, our procedure inherits the thresholds, detection method, and removal decisions documented in the original works. This also ensures that we do not introduce additional class-specific bias beyond what these curated test sets already corrected for.
> >
> >
> > **"Can the authors include a controlled comparison showing accuracy changes when removing top-scored versus random samples under both the original and revised scoring methods?"**
> >
> > We thank the reviewer for this insightful suggestion. In response, we reran the experiments using the original Feldman–Zhang method (e.g., use sample rate of 0.7 using the original scoring method and the threshold of 0.25). This acts as our baseline. Next, we modify only one component at a time (e.g., increase the sample rate to 0.95). We include the results in Table 1 (in the newly created Section 5: Ablation Study) also shown below:
> >
> > | Configuration                         | Utility | Improvement Over Baseline | p-value            | Significant |
> > |--------------------------------------|---------|----------------------------|--------------------|-------------|
> > | Original FZ (Baseline)               | -1.26   | --                         | --                 | --          |
> > | + Fractional Difference Definition   | -1.17   | +0.09                      | 0.6064             | No          |
> > | + De-duplicated Test Set             | -0.91   | +0.35                      | 0.0971             | No          |
> > | + Sampling Rate 0.95                 | -0.44   | +0.82                      | 9.21×10⁻⁴         | Yes         |
> > | + Principled Thresholding            | -0.04   | +1.22                      | 5.54×10⁻⁹         | Yes         |
> > | **All Four Components Combined**      | **+0.30** | **+1.56**                 | **1.03×10⁻²**     | **Yes**     |
> >
> >
> > There are a few key takeaways here:
> > 1. **Role of the Scoring Method:** Our results show that changing the scoring method does not have a statiscally significant impact on utlity. Eventhough utility increases from –1.26% (FZ Baseline) to –1.17%, this is not a statistically significant gain (p > 0.05). This is likely due to the fact that we are using the arbritary threhsold of 0.25 in both these experiments. Such a low threshold likely introduces false positives, wiping out the gains achieved by improving the scoring method.
> >
> > 2. **No single factor alone is sufficient:** Sampling rate 0.95 contributes +0.82 improvement, but still leaves utility negative. In contrast, principled thresholding contributes an additional +1.22 improvement and is statistically significant at p < 10⁻⁸. Fractional difference and de-duplication provide smaller (and individually non-significant) improvements, but their combined effect is necessary for the final stability of the score. When all four factors are applied jointly, the utility becomes positive (+0.30), reversing the sign of the effect and yielding statistically significant improvements.
> >
> > 3. **Thresholding appears to be a "bottleneck fix."** Principled thresholding alone accounts for +1.22 of the +1.56 total improvement (about 78%). This indicates that the use of the arbitrary thresholding step in FZ, where they choose the 0.25 threshold for memorization without any justification, is the primary source of the observed drop in utility. This is likely because unprincipled thresholding can cause non-memorized points to be misclassified as memorized, thereby removing samples that actually support generalization.

---

### Official Review · Reviewer_LVvf · 2025-11-01

**Soundness:** 2
**Presentation:** 3
**Contribution:** 2
**Rating:** 4
**Confidence:** 4

**Summary:**

This paper revisits Feldman & Zhang (FZ, 2020) and argues that their core claim “memorization is necessary for generalization” is an artifact of methodology. The authors identify four issues in FZ’s pipeline: (1) sampling bias from training on small subsets, (2) a high false-positive definition of “memorization,” (3) unprincipled thresholding, and (4) train-test leakage. They propose fixes: train at higher sampling rates (near LOO), replace the absolute difference score with a fractional difference score, adopt a null-distribution threshold, and evaluate on deduplicated test sets. Under these changes, they report that removing “memorized” points never hurts and often improves test accuracy, especially on CIFAR-100 and ImageNet.

**Strengths:**

1. Clear diagnosis of pitfalls. The four highlighted issues (sampling bias, high FPR, arbitrary thresholding, leakage) are well-motivated and relevant to the broader literature.
2. Better detection criterion. The fractional-difference score is a sensible refinement; the paper evaluates detection quality using FPR@95% TPR, showing reductions in false positives relative to the original definition.
3. Leakage awareness. The paper documents near-duplicates and evaluates on non-leaky test sets (CIFAR-no-dups, ImageNet-V2), which strengthens claims about generalization.

**Weaknesses:**

1. The paper challenges a seminal result (Feldman–Zhang, “FZ”) but doesn’t bring the mountain of evidence needed to overturn it; the current experiments and analysis aren’t enough for a contrary claim.

2. Fact-level imprecision: the manuscript says FZ used only a 0.7 subset. In reality, FZ released CIFAR-100 models spanning 0.1–0.9, susbet for CIFAR100 with 0.7 only for ImageNet.

3. Sampling-bias critique undercut by scale: authors claim sample bias yet train only ~50 models and provide no ablation on number of modes. FZ trained about 2,000 ImageNet and 10,000 CIFAR-100 models (for each subsampling ratio on cifar100) without matching that scale, tail effects and variance aren’t convincingly addressed.

4. Underuse of existing resources: instead of training models, the paper could directly use FZ’s released models to replicate and stress-test claims, removing implementation confounds. Using FZ models to prove the opposite would be the STRONGEST case for the claims of the paper, this I feel like this is a missed opportunity given the scale of model trained by Feldman–Zhang which is also publicly available.

5. Evaluation too narrow: the new metric is interesting, but the paper mostly reports FPR@95% TPR. The paper should potentially also include full ROC/PR curves, AUROC, and AUPR, with confidence intervals and class-wise breakdowns.

6. Weak theoretical engagement: the rebuttal to FZ’s long-tail theory is basically two statements of intuition (see lines 277–280) with no rigorous counter-argument or alternative theory. The experimental results are solid, but purely empirical; they don’t actually refute the theoretical result.

Minor:
Citation/presentation problems: at least one missing reference (around line 330 and 332)

**Questions:**

See weaknesses, but my general stance is that I do believe the authors are on to something but when making contrary claims to well established results the mountain of evidence to prove the contrary is missing from the paper. The use of FZ models to prove the contrary to FZ results would be the best argument for the paper.

---

> ### Author Response · Authors · 2025-11-23
>
> **"Fact-level imprecision: the manuscript says FZ used only a 0.7 subset. In reality, FZ released CIFAR-100 models spanning 0.1–0.9, susbet for CIFAR100 with 0.7 only for ImageNet."**
>
>
>
>
>
> Thank you for pointing out this detail. While it is true that the authors later released CIFAR-100 models trained with a range of sampling rates, their *published* experimental results, in the main text, tables, and figures used to support their conclusions, are based exclusively on the 0.7 rate. This includes all memorization estimates and the core empirical claims made in the original paper. Because our contribution is to re-evaluate the methodology underlying these published results, we intentionally matched the configuration used in their experiments rather than the broader set of models released afterward. Our goal is to challenge the conclusions drawn in the original manuscript, and therefore we focused on the specific sampling rate that their findings rely on.
>
>
>
>
>
>
>
> **"Sampling-bias critique undercut by scale: authors claim sample bias yet train only ~50 models and provide no ablation on number of modes..."**
>
>
>
> Thank you for raising this concern. We apologize for the confusion in the earlier draft.
>
>
>
> 1. Our experiments do not use only 50 models for calculating memorization scores, we use 500 for CIFAR-10/100 and 150 for Imagenet (Appendix A.1).
>
>
> 2. To ascertain these ensemble sizes produce stable memorization scores, we conducted a sweep over the number of models (now present in Figure-9). We find that the Spearman correlation between memorization scores and ens stabilizes at roughly 300 models for Cifar10/100 and 150 for Imagenet. Beyond this point, the memorization scores do not change. Nevertheless, for completeness and to match the spirit of Feldman & Zhang’s large-scale setup, we trained an additional 200 models, bringing the total to 500 (for CIFAR10/100). Even though the imagenet memorization scores converge with 150 models, we are actively training more imagenet models to raise the total to 500 as well. We hope to update the paper before the end of the discussion period.
>
>
>
> 3. Importantly, if the number of models were infact not sufficient, we would have seen an increase noise in the score estimates, which would manifest as spurious memorized points and **degraded** test accuracy after removal. Instead, we observe the opposite: removing high-score points consistently improves generalization. This provides strong evidence that our results are not an artifact of insufficient models but rather hold robustly even at this scale.
>
>
>
> 4. We appreciate the point regarding the Ablation modes. We ran an ablation study (results now shown in Table 1) to observe which one of the four componenets are necessary. We found that: (a) No single factor is sufficient to make utility positive. (b) The four factors do not contribute equally. Order of impact on utility (highest to lowest) Principled Threshold --> Sampling Rate of 0.95 --> Data Leakge Removal --> Fractional Difference Definition. (c) Principled Threshold is the bottle neck fix, contributing the largest improvement in utility.
>
>
>
>
>
> **"Underuse of existing resources: instead of training models, the paper could directly use FZ’s released models..."**
>
>
>
> Thank you for this thoughtful suggestion. We initially considered using the publicly released Feldman–Zhang models to replicate and stress-test their claims. However, we ultimately decided against this approach for two reasons.
>
> 1. The released models cover limited set of architectures (RESNET50 and InceptionNet). A core part of our contribution is demonstrating that the identified issues persist across datasets and model families (CIFAR-10, CIFAR-100, ImageNet; ResNet, VGG, MobileNet). Restricting our analysis to the FZ model collection would prevent us from making any claims about robustness or generality.
>
>
>
> 2. We found that memorization scores are sensitive to the underlying GPU hardware due to well-documented differences in mixed-precision arithmetic across GPU generations. Feldman–Zhang did not report the hardware used to train their released checkpoints. Using their models while training our own comparison models on different hardware would introduce a confound that directly affects the marginal-utility experiments, which rely on consistency across models. To avoid such hardware-induced bias, we chose to train all models ourselves under a controlled and uniform setup.
>
>
>
> This means that the conclusions we draw are attributable to the memorization methodology rather than implementation differences or hardware mismatch.

---

> ### Author Response · Authors · 2025-11-23
>
> **"Evaluation too narrow: the new metric is interesting, but the paper mostly reports FPR@95% TPR..."**
>
>
>
> We thank the reviewer for this helpful suggestion. In the revised version, we have expanded our evaluation beyond FPR@95% TPR: we now report AUROC scores and include full ROC and PR curves for all datasets in Figure 7. Our results show that across all settings, our definition consistently outperforms the original memorization score, achieving higher AUROC and AUPR on CIFAR-10, CIFAR-100, and ImageNet.
>
> Class-wise breakdowns are more challenging to display in the main text due to the large number of classes in CIFAR-100 and ImageNet. If the reviewer wants this reported in a particular format, please let us know.
>
>
>
>
>
> **"Weak theoretical engagement: the rebuttal to FZ’s long-tail theory is basically two statements of intuition"**
>
>
>
> Thank you for highlighting this point. We agree that our work does not provide a new theoretical framework to replace the long-tail explanation proposed by Feldman & Zhang. Our contribution is primarily empirical: we identify methodological issues in how memorization is measured and show, through extensive experiments, that the empirical phenomena reported in FZ do not hold once these issues are addressed.
>
>
>
> Developing a full theoretical alternative is indeed valuable but is beyond the scope of our current work, which focuses on correcting the empirical measurement pipeline and reevaluating the conclusions drawn from it. We can make this explicit in the revision and include a dedicated limitations section noting that our results  do not provide a theoretical account of memorization dynamics.

---

### Official Review · Reviewer_KoMs · 2025-11-02

**Soundness:** 3
**Presentation:** 3
**Contribution:** 2
**Rating:** 4
**Confidence:** 4

**Summary:**

This paper challenges the influential claim by Feldman & Zhang (2020) that memorization is necessary for generalization. The authors posit that the conclusion of the original work is founded upon four flaws: (1) sampling bias in the memorization score approximation algorithm, (2) a high false positive rate in the original definition of memorization, (3) an unprincipled and arbitrary threshold for identifying memorized points, and (4) data leakage between the training and test sets. The paper introduces four corresponding modifications: (1) using a high sampling rate to approximate the leave-one-out (LOO) setting, (2) a new "fractional difference" definition for the memorization score, (3) a statistically principled threshold based on a null distribution, and (4) the use of curated test sets designed to prevent train-test overlap. Upon applying these corrections, the authors claim the original conclusion by Feldman & Zhang (2020) must be reversed.

**Strengths:**

* I appreciate the critical view adopted by this paper. The paper identifies several legitimate methodological weaknesses in the work of Feldman & Zhang (2020), and as the authors state potentially in many other ML papers. The proposed solutions, particularly the use of a null distribution for thresholding and the identification of data leakage, are theoretically sound.

**Weaknesses:**

* It is important to distinguish between truly hard samples that must be memorized and those that are memorized due to their unfavorable properties (e.g., being noisy/outlier or duplicate). It seems the example used in section 3.2 looks at the memorization phenomenon from a purely numerical perspective. $x_o$ could be a noisy sample hence having a high memorization score, indicating the model is sensitive to its presence while $x_i$ could be a truly hard example. If one agrees with this premise, the proposed score in eq 2 inflates the score of noisy samples and mark them as memorized while suppress that of truly hard samples. This could be misleading and in fact could be the underlying factor behind the conclusion of the paper: if truly hard samples are not marked as memorized and only noisy ones are marked as such, given that the latter have nothing to do with the test data, are not needed for generalization. Building upon this, I suggest the authors to use their proposed score in eq 2 for dataset cleaning, i.e. measure its utility for identifying noisy and duplicate ones.

* The experiments lack some important details. Please see the Question section.

**Questions:**

* The paper claims the denominator $Pr_{in} + Pr_{out}$ in eq 2 was chosen for "numerical stability" and cites "Wu & Baleanu (2018)". But it seems this is a paper on dynamical systems and fractional calculus, which has no discernible relevance to the statistical properties of estimator of the memorization score proposed in eq 2. Can you comment which specific result from Wu & Baleanu (2018) substantiate your claim of numerical stability?

* Section 4.1 states, "To evaluate whether memorization is necessary for generalization, we trained 50 models for each of the two conditions". However, Appendix A states, "During marginal utility, we train 10 models in for each threshold". Is this a typo?

* The caption for Table 2 states, "CIFAR values are scaled by 100, while ImageNet values are scaled by 200 for readability". Do the authors mean accuracies by "value"? If so, this note is incoherent. If the "Mem Remove Acc (%)" for ImageNet (e.g., 12.26) were scaled by 200, the true accuracy would be $12.26 / 200 = 0.0613\%$, which is nonsensical.

* The p-values reported in Table 2 are strange. For ImageNet/VGG19, the p-value is $6.06 \times 10^{-45}$. For ImageNet/ResNet 18, it is $1.35 \times 10^{-37}$. Achieving such astronomical levels from only 50 runs of a deep learning experiments implies a variance that is effectively zero. Is this due to using a high sampling rate of 0.95?

* I'd like to see the results reported in Figure 5 and Table 2 using the original FZ score with sampling rate 0.95 as opposed to 0.7 used originally by Feldman and Zhang and see how it compares with the proposed approach. Overall, the ablation study needs to be improved to see whether all of the four components are essential or just one, e.g. high sampling rate is adequate. I am willing to increase my score depending on the authors' response to this particular question.

---

> ### Author Response · Authors · 2025-11-23
>
> ## Ablation Experiments:
>
> We thank the reviewer for this insightful suggestion. In response, we reran the experiments using the original Feldman–Zhang method (e.g., use sample rate of 0.7, using the original definition and threshold of 0.25). This acts as our baseline. Next, we modify only one component at a time (e.g., increase the sample rate to 0.95). We include the results in Table 1 (in the newly created Section 5: Ablation Study) also shown below:
>
> | Configuration                         | Utility | Improvement Over Baseline | p-value            | Significant |
> |--------------------------------------|---------|----------------------------|--------------------|-------------|
> | Original FZ (Baseline)               | -1.26   | --                         | --                 | --          |
> | + Fractional Difference Definition   | -1.17   | +0.09                      | 0.6064             | No          |
> | + De-duplicated Test Set             | -0.91   | +0.35                      | 0.0971             | No          |
> | + Sampling Rate 0.95                 | -0.44   | +0.82                      | 9.21×10⁻⁴         | Yes         |
> | + Principled Thresholding            | -0.04   | +1.22                      | 5.54×10⁻⁹         | Yes         |
> | **All Four Components Combined**      | **+0.30** | **+1.56**                 | **1.03×10⁻²**     | **Yes**     |
>
>
> There are a few key takeaways here:
> 1. **Role of Sample Rate:** Increasing the sampling rate does improve utility over the FZ baseline, but only marginally. Utility increases from –1.26% (FZ Baseline) to –0.44%, a statistically significant gain (p < 10⁻³). However, this modification is not sufficient to recover the correct behavior. Even at a high sampling rate, there is a net *decrease* in test accuracy after removal.
>
> 2. **No single factor alone is sufficient:** Sampling rate 0.95 contributes +0.82 improvement, but still leaves utility negative. In contrast, principled thresholding contributes an additional +1.22 improvement and is statistically significant at p < 10⁻⁸. Fractional difference and de-duplication provide smaller (and individually non-significant) improvements, but their combined effect is necessary for the final stability of the score. When all four factors are applied jointly, the utility becomes positive (+0.30), reversing the sign of the effect and yielding statistically significant improvement over both the original method and the “sampling-rate” variant.
>
> 3. **Thresholding appears to be a "bottleneck fix."** Principled thresholding alone accounts for +1.22 of the +1.56 total improvement (about 78%). This indicates that the use of the arbitrary thresholding step in FZ, where they choose the 0.25 threshold for memorization without any justification, is the primary source of the observed drop in utility. This is likely because unprincipled thresholding can cause non-memorized points to be misclassified as memorized, thereby removing samples that actually support generalization.
>
>
> In short, the additional experiment requested by the reviewer confirms the core conclusion of our work: a high sampling rate alone is not adequate, and the original method fails for multiple independent reasons. Only the complete set of corrections (sampling bias, fractional-difference definition, principled thresholding, and leakage removal) reveals the true behavior of memorized points.
>
>
>
> ## The p-values reported in Table 2 are strange.
> We appreciate the reviewer’s careful attention to the p-values. The extremely small values are not caused by the high sampling rate of 0.95, nor do they imply that the variance of the deep-learning runs is “effectively zero.” Instead, they are caused by the the large difference in utility between “memorized removal” and “random removal”. In other words, the mean difference across runs is multiple percentage points, while the run-to-run standard deviation within each condition is very small. The p-values do not indicate degenerate variance or instability. They arise because the memorized vs. random removal gap is large and stable across runs. The variance is small, but not zero, and as a result, the p-values remain very small.

---

> > ### Author Response · Authors · 2025-11-23
> >
> > ## It is important to distinguish between truly hard samples
> > 1. The reviewer raises an important distinction between truly hard samples and noisy or outlier samples. However, the premise that the original difference score correctly favors hard samples is not supported by its behavior. Lets consider a starker example than the one in Section 3.2. Under the original definition, a point x_i that the model predicts with very high confidence (e.g., whose IN and OUT predictions differ from 100% to 99%) receives the same absolute memorization score (a 1-point change) as a noisy or mislabeled point x_o whose prediction oscillates between 1% (IN) and 0% (OUT).
> > These two situations are fundamentally different: x_i is extremely stable with respect to model. It can not at all be considered a hard sample.  On the other hand, x_o reflects high instability and is typically associated with noise or corruption. Because the original score relies on an unnormalized difference, it cannot distinguish small fluctuations around a confident prediction from large relative instabilities near low-confidence predictions.
> >
> > 2. We have added the images of the samples identified as memorized under the null distribution to the supplementary material. Notably, these high-score images appear visually typical and representative of their respective classes (i.e., this indicates these images are not just outliers, but hard samples). Specifically, they do not exhibit noise, distortions, or outlier-like patterns. This supports our argument that the score is not just simply identifying outliers, but can also identify hard samples.
> >
> >
> > Our fractional-difference score in Eq. (2) does not suppress hard samples. Instead, it suppresses negligible high-confidence changes while amplifying large relative changes, thereby separating the generalizable easy samples from noisy or corrupted ones.
> >
> > ## Other errors
> > We thank the reviewer for pointing out the other textual errors in the manuscript. We have gone ahead and modified the text to remove the confusion. We removed the incorrect citation, corrected the number of models used for training, and fixed the table caption.

---

### Meta-Review · Area_Chair_tTsP · 2025-12-27

**Summary:**

The paper challenges conclusions made in by Feldman & Zhang (FZ). In essence, FZ claims that removing memorized points degrades generalization (even more so than of random ones) and provided a theoretical explanation.

The paper makes a generally convincing claim that FZ had several issues. In particular: data leakage (public vision benchmarks such as CIFAR10 are known to have some degree of near duplicates) and thresholding

Authors have rebutted successfuly most of comments made by Reviewers. In particular, they added an important ablation during the rebuttal phase which shows that thresholding was the most important change.

The argument for why one shouldn’t try to reuse released models (at least showing results on those models would be helpful) wasn’t fully convincing. Authors also did not agree to release code, which I believe is essential for such a paper.

It has been shown in multiple works that removing memorized examples can improve generalization (see e.g. https://arxiv.org/abs/2003.10647). It is also standard practice to remove low quality examples from pretraining dataset for training LLMs. Therefore, the community is generally aware that memorization is hurting generalization or at the best is neutral, in many settings. From this perspective, the contribution is limited to the very particular claims made in the paper, rather than the broad issue of “memorization is needed for generalization”.

All in all, I am recommeding rejection at this stage. The paper main claim is well supported by the paper. However, the broader impact is potentially limited by (1) the fact that the community is already aware that memorization hurts generalization in many cases, and (2) by the limitations in reproducibility (lack of released code, not comparing using FZ models).

**Reviewer Concerns:**

Among the most important comments made:

- **Reviewer LVvf**, **Reviewer XWnb** argued the authors used only ~50 models to estimate memorization scores, arguing this was insufficient compared to FZ’s thousands of models.
- **Reviewer KoMs and Reviewer AsrV** requested a controlled comparison to isolate which of the four proposed fixes mattered most.
- Reviewer **LVvf** suggested comparing to models released by FZ
- Reviewer XWnb was concerned that the paper doesn’t release code

**Reviewer Scores:**

All reviewers gave the paper score 4 (weak reject).

---

### Decision · Program_Chairs · 2026-01-26

Reject